# Sexual Health and Psychological Well-Being of Women: A Systematic Review

**DOI:** 10.3390/healthcare11233025

**Published:** 2023-11-23

**Authors:** Ana Isabel Arcos-Romero, Cristobal Calvillo

**Affiliations:** 1Department of Psychology, Universidad Loyola Andalucía, 41704 Seville, Spain; 2Department of Health Behavior and Health Education, University of Arkansas for Medical Sciences, Little Rock, AR 72205, USA; cfcalvillomartinez@uams.edu

**Keywords:** sexual health, psychological well-being, women, systematic review

## Abstract

(1) Background: Psychological well-being (PWB) and female sexual health are two important areas for women’s quality of life and research, and they are closely related. The aim of this study was to conduct a systematic review of the existing literature to explore the association between PWB and sexual health in women. (2) Methods: This review was carried out following the PRISMA checklist. The inclusion criteria were studies with samples of adult women that evaluated and associated sexual functioning and psychological well-being. Scientific articles were identified on Web of Science, Scopus, EBSCO (PsycInfo, PsycArticles, and Psicodoc), ProQuest, and PubMed. The search was limited to years between 2010 and 2023. The methodological quality of the studies was assessed using the Quality Assessment Tool for Observational Cohort and Cross-Sectional Studies (QATOCCS). (3) Results: 14 selected articles were analyzed, in which population samples and variables related to psychological and sexual health were examined. In total, 42.9% of the studies included clinical samples, 71.4% focused on anxiety and depression as the main psychological variables, and 50% examined female sexual functioning as a sexual health variable. (4) Conclusions: This review provides more up-to-date information about valuable insights into the possible determinants of female sexual health. An association between PWB and female sexual health has been demonstrated.

## 1. Introduction

Psychological well-being (PWB) and female sexual health are two important areas for women’s quality of life and research, and it is well-known that there is a close relationship between both aspects. PWB is commonly characterized as reaching one’s full potential and is linked to meaningful (eudaimonic) wellness attitudes [1], and it embraces different aspects such as autonomy, environmental mastery, personal growth, positive relations with others, purpose in life, and self-acceptance [2,3]. Sexual health, as mentioned by the World Health Organization (WHO) [4], refers to “a state of physical, emotional, mental, and social well-being in relation to sexuality” (p. 5). It emphasizes the importance of a positive and respectful approach to ensure pleasurable and safe sexual experiences [4]. When it comes to women, female sexual health encompasses various physical, emotional, and social factors that influence their sexual experiences and overall quality of life. Research consistently shows associations between PWB and sexual health outcomes across diverse cultures [5]. Some examples of these associations have shown that positive PWB or better mental health, including high self-esteem, life satisfaction, and lower levels of depression and anxiety, are associated with improved sexual functioning and satisfaction in women [6,7,8]. For example, young women with higher self-esteem reported higher sexual functioning, greater ability and intensity of orgasm, and higher sexual satisfaction [7]. Conversely, negative PWB and/or mental health issues, such as depression and anxiety, have been found to be associated with sexual dysfunction, decreased sexual desire, and difficulties with sexual arousal and orgasm [5,9,10,11]. Lastly, it has been observed that psychological well-being is addressed broadly with constructs of psychological health, mental health, mental well-being, or absence of mental diagnosis. Based on Woloski-Wruble et al. [8], mental (or psychological) well-being is a cognitive and physical functional capacity, and an emotional and social well-being, all related to an active engagement with life.

Along the same line, it is important to acknowledge that female sexual dysfunction (FSD) is a complex issue influenced by various factors, including biological, psychological, and interpersonal aspects [12]. Psychological factors play a significant role in FSD, as evidenced by studies in women with diabetes, in which a significant predictor for sexual dysfunction was depression [OR 6, 95% CI for estimated odds ratio 2.0–18.0] [13] and there was an association between sexual dysfunction and psychological distress in women with depression [14]. Furthermore, sexual dysfunction—in general—significantly impacts community health, particularly affecting women. Several international studies have provided compelling evidence regarding the prevalence of sexual dysfunction. For instance, in the United States, 43% of women reported experiencing sexual dysfunction, compared to 31% of men [15]; in Turkey, the prevalence of FSD was found to be 53.2% [16]; in China, it reached 60% [17]; Brazil exhibited a prevalence ranging from 13.3% to 79.3% among women [18]; and, lastly, in Latin America, a study involving 5391 individuals found that 56.8% of sexually active women aged 40–59 experienced sexual dysfunction [19]. To promote comprehensive sexual well-being for women is crucial for researchers and healthcare professionals to acknowledge and address the intricate relationship between PWB and sexual health outcomes. By understanding the interplay between these factors, healthcare providers can develop more effective strategies to support women in achieving optimal sexual health and overall well-being.

Regarding a different matter, given the fact that systematic reviews are considered to be “one of the highest forms of research evidence” [20] (p. 199), and “answer an empirical question based on a minimally biased appraisal of all the relevant empirical studies” [21] (p. 121), conducting a systematic review of PWB and female sexual health could provide a stronger and more reliable understanding by combining findings from multiple studies. Also, this review could identify gaps, inform future research, and promote overall well-being and quality of life for women. Despite the significance of this topic, more research is needed to identify the risk factors for sexual dysfunction and better understand the interaction between PWB and female sexual health. Lastly, an examination of recent studies with certain populations and information on the most-used measures that assess women’s sexual and psychological health are issues that need to be addressed in research.

In summary, it is known that human sexuality is an important part of peoples’ lives and well-being [22]. The link between psychological well-being and female sexual health is crucial to women’s quality of life. This work started from the following research question: How is psychological well-being related to sexual health in women? We hypothesized that better psychological well-being will be related to better female sexual health. As aware as we are, a systematic review of the scientific literature specifically focused on these two topics in women does not exist. Thus, the aim of this study is to conduct a comprehensive review of the existing scientific literature to explore the association between PWB and sexual health in women.

## 2. Materials and Methods

### 2.1. Study Design

This systematic review was carried out following the PRISMA statement [23]. Due to no human participants being involved, there were no requirements for an ethical review of this work.

### 2.2. Eligibility Criteria

The inclusion criteria for the review were as follows:Studies with samples of adult women from both clinical and non-clinical contexts.Studies that evaluated sexual health and PWB variables.Studies that associated sexual health and PWB variables.

The exclusion criteria were documents with other methodological designs (systematic, narrative, bibliographic reviews, or non-relational qualitative studies), other types of documents (doctoral theses or other academic works), studies in a language other than English or Spanish, studies with female samples under 18 years of age such as adolescents or girls, and studies that did not include an analysis of sexual health or sexual functioning and psychological well-being variables and their association. Studies in this review were grouped for the syntheses.

### 2.3. Information Source

The studies were identified on Web of Science, Scopus, EBSCO (PsycInfo, PsycArticles, and Psicodoc), ProQuest, and PubMed. The dates when each source was last searched were between 1 February 2023 and 30 April 2023.

### 2.4. Search Strategies

The search strategy was (“Sexual health” AND “Psychological well-being” AND (“women” OR “female”)). We used the filters and limits for the publication years between 2010 and 2023, language in English or Spanish, and scientific articles in document types.

### 2.5. Selection Process

Working independently, two reviewers screened each record retrieved to decide whether they met the inclusion criteria of this review. No automation tools were used in the process.

### 2.6. Data Collection Process

To guarantee the objectivity and rigor of the results, two reviewers collected data from each database by working independently. Then, they pooled and confirmed the data. Based on the inclusion criteria, a screening of the titles was conducted. Studies that clearly did not respond to the review objectives and duplicates were excluded. The two reviewers screened the abstracts separately. Finally, the most relevant studies were full text read. No automation tools were used in the process.

### 2.7. Data Items

Data from studies with samples of adult women (clinical and non-clinical) that evaluated sexual health and psychological well-being variables and associated them were considered.

### 2.8. Study Risk of Bias Assessment

To assess the risk of bias in the selection of the included studies, two reviewers evaluated each study by working independently. After reading the articles that met the inclusion criteria, their methodological quality was assessed using the Quality Assessment Tool for Observational Cohort and Cross-Sectional Studies (QATOCCS) [24]. This tool ensures scientific quality and methodological transparency by evaluating aspects such as the clarity of the research question and the definition of the study population through a checklist of requirements (e.g., “Was the research question or objective in this paper clearly stated?” or “Was the study population clearly specified and defined?”). The rigorous evaluation ensured the studies’ robustness and the results’ reliability. For the current analysis, the two reviewers independently read and rated each of the selected studies across the QATOCCS criteria. Any discrepancies in ratings were discussed until a consensus was reached. This comprehensive quality assessment ensured that only methodologically robust studies with reliable psychometrics and minimal bias were included, strengthening the reliability of the results. The QATOCCS provided a standardized and transparent framework for critically evaluating the scientific rigor of all the studies under consideration. Furthermore, the psychometric properties of the measurement instruments used in each study were considered.

### 2.9. Synthesis Methods

We tabulated, for better visualization, the authors and year of publication, samples, type of samples, list of psychological well-being variables, list of sexual health variables, measurement instruments, main results about how the observed variables were associated (type of results and type of association), and the understanding of PWB of each article.

## 3. Results

The search yielded a total of 174 articles. After analyzing them to determine their compliance with the inclusion criteria and assessing their quality, a final set of 14 articles remained for further analysis (see Figure 1). Thus, in the present systematic review study, 14 selected articles were analyzed, in which population samples and variables related to psychological well-being and sexual health were examined. Regarding the former, it was found that 42.9% of the studies used clinical samples as the study population [25,26,27,28,29,30], while 35.7% used representative samples of the general population [31,32,33,34,35]. Only three studies (21.4%) used both types of samples, clinical and general populations [36,37,38]. Table 1 presents the reviewed articles about sexual health and psychological well-being.

Secondly, regarding variables related to psychological well-being, it was observed that anxiety and depression were the most investigated constructs. In total, 71.4% of the studies examined these constructs individually, in combination, or with other psychological health constructs. Only four studies (28.6%) of the reviewed articles investigated variables other than anxiety or depression, such as sense of responsibility [31], attention deficit hyperactivity disorder [37], constructs related to objectification [33], and other aspects of psychological health [30]. In relation to sexual health variables, 50% of the studies exclusively evaluated variables linked to sexual functioning, while two articles (14.3%) analyzed aspects other than sexual functioning but in relation to sexual health [28,33]. Five articles (35.7%) examined variables associated with sexual functioning along with other variables related to sexual health [26,27,29,37,38]. For example, Fogh et al. [26] reported odds ratios indicating an increasing probability of sexual dysfunction with more/worse depressive symptoms.

Concerning the measures used to assess variables related to psychological well-being and sexual health, it was found that standardized scales were used in most of the reviewed studies. Most of the measures for assessing variables related to psychological aspects focused on anxiety or depression. To measure anxiety exclusively, the Florida Shock Anxiety Scale (FSAS) by Ford et al. [39] and the Generalized Anxiety Disorder Scale (GAD-7) by Löwe et al. [40] were employed. As for depression, the Beck Depression Inventory (BDI) [41], the Beck Depression Inventory-II (BDI-II) [42], and the Beck Depression Inventory Primary Care (BDI-PC) [43] were used. Only three articles (21.4%) employed measures that assessed both anxiety and depression, such as the Hospital Anxiety and Depression Scale (HADS) by Zigmond and Snaith [44], the Brief Symptom Inventory (BSI) by Derogatis [45], and an ad hoc measure that evaluated general aspects of psychological health. Considering the measures used to assess variables related to sexual health, the Female Sexual Function Index (FSFI) developed by Rosen et al. [46] was the most used in nine studies (64.3%), being the predominant scale in this review. Of the nine studies, four studies exclusively used the FSFI, while the other five studies used the FSFI along with other scales to assess various constructs related to sexual health. It is worth mentioning that, in the study by Vedovo et al. [34], both the FSFI and the Operated Male to Female Sexual Function Index (OMtFSI) [47] were used, with the latter being the first scale designed to assess sexual functioning in trans women. Other scales used to assess sexual functioning or related aspects included the Female Sexual Function Questionnaire-2 (FSM-2; for its acronym in Spanish) designed by Sánchez-Sánchez et al. [48], the Global Measure of Sexual Satisfaction (GMSEX) by Lawrance et al. [49], an ad hoc questionnaire used by Dubin et al. [31] to assess desire and satisfaction, and a Female Orgasmometer, a single-item Likert scale derived from the Visual Analog Scale for Pain [50], which was used in the study by Mollaioli et al. [32]. Other scales that did not directly assess sexual functioning but rather other aspects of sexual health were used such as the Sexual Distress Scale (SDS) created by Santos-Iglesias et al. [51], the Sexual Complaint Screener-Women (SCS-W) developed by the International Society of Sexual Medicine (ISSM) [52], the Sexual Risk Survey (SRS) by Turchik and Garske [53], and the Interpersonal Sexual Objectification Scale (ISOS) by Kozee et al. [54], among others (see Table 1).

Finally, the results of the reviewed articles were categorized according to their focus, whether contextual, medical, or psychological. Seven articles (50%) presented results that combined two of these categories. Of these, five were categorized as contextual–psychological, with risk factors influencing psychological and sexual health variables. They explored factors related to the COVID-19 pandemic, social stigmas towards women, and postponed fertility treatments [25,28,32,33,35]. For example, Mollaioli et al. [32] reported significant odds ratios about the association between cessation of sexual activity during lockdown and a higher risk of developing both anxiety and depression. The other two articles were categorized as medical–psychological, focusing on risk factors related to ADHD and hypoactive sexual desire diagnosis, respectively [30,37]. Additionally, another four articles (28.6%) centered only on medical outcomes, addressing factors related to cancer, menopause, and pelvic–genital pain [26,27,29,38]. Two articles (14.3%) presented exclusively psychological outcomes, with anxiety and depression as the risk factors [35,36]. Finally, only one article (7.1%) addressed contextual outcomes, specifically the partner context, by examining couple’s problems caused by the partner’s erectile dysfunction [31].

## 4. Discussion

The aim of this study was to carry out a systematic review of the association between PWB and female sexual health. In general, the findings show that recent research has focused on women of clinical samples and considers anxiety and depression as the most important PWB variables, while general sexual functioning is the most relevant sexual variable.

This recent scientific literature that examined the association between psychological well-being and sexual health has shown great interest in women diagnosed with certain health problems. Therefore, the majority of the studies focused on women with specific medical diagnoses when evaluating the link between PWB and sexual health. These medical diagnosis topics included issues related to congenital heart disease with implantable cardioverter-defibrillators, and breast, anal, or rectal cancer. For example, results have shown that, in women with congenital heart disease, depression and anxiety are negatively associated with worse sexual functioning [36]. Some psychological disorders like Attention-Deficit/Hyperactivity Disorder (ADHD) and opioid use disorder are also considered. For example, women with ADHD reported more hypersexual behaviors [37]. Also, the literature has considered sexual and reproductive health problems such as Persistent Genital Arousal Disorder/Genito-pelvic Dysesthesia (PGAD/GPD), Hypoactive Sexual Desire Disorder (HSDD), and fertility. In this line, individuals with PGAD/GPD symptoms indicated lower relationship satisfaction and sexual satisfaction and greater sexual distress in comparison to a control group [38]. All this information could reveal a greater interest in medical aspects rather than psychological or sexual ones when assessing clinical samples.

Anxiety and depression are the most investigated variables in psychological well-being when the objective is also examining the sexual health of the women in the reviewed studies. Philip et al. [29] analyzed psychological well-being through anxiety and depression and found that anxiety is negatively associated with desire, arousal, orgasm, and satisfaction, while depression is negatively associated with arousal and sexual satisfaction in post-treatment anal or rectal cancer survivors. Mooney et al. [38] addressed psychological well-being as the absence or presence of depressive or anxiety symptoms and demonstrated that women with PGAD/GPD report more symptoms of depression and anxiety. Building upon this finding, Mollaioli et al. [32] found in their study that sexual functioning has a marker and predictive role in psychological well-being—which is assessed with anxiety and depression measures—and further established the beneficial impact of sexual activity in safeguarding against psychological distress, promoting relational well-being, and enhancing sexual health. It is noteworthy that a significant proportion of the examined articles predominantly framed the concept of psychological well-being with a clinical paradigm, with a particular emphasis on the identification and analysis of symptoms associated with mental health conditions, notably anxiety and depression. Furthermore, a considerable portion of these investigations operationalized psychological well-being as the absence of such deleterious symptoms, without extensively considering the positive dimensions of mental health. This matter may impose constraints on the comprehensive exploration of the intricate intersections between various aspects of mental health and sexuality in women. Notably, there is little research that has incorporated assessments of constructs such as self-esteem, self-efficacy, resilience, and other strengths-based factors. The inclusion of a broader spectrum of positive psychology perspectives, encompassing the evaluation of emotional, social, and psychological resources and competencies, could significantly enhance the depth of analysis concerning the interplay between diverse dimensions of mental health and their associations with sexual health outcomes in women.

Moreover, research examining female sexual health has primarily focused on sexual functioning, often overlooking other critical aspects like sexual behaviors, attitudes, gender roles, and the prevention of sexual and reproductive infections or diseases, unwanted pregnancies, or sexual violence. As a result of this focus, and in terms of assessment, the most used measurement instrument for evaluating sexual functioning in women is the FSFI [46]. In this line, the quality assessment of women’s sexuality requires standardized validated scales encompassing multidimensional constructs. Beyond the Female Sexual Function Index, other validated questionnaires can provide an assessment of sexual functioning across genders. Two examples of valid measurement instruments are the Massachusetts General Hospital Sexual Functioning Questionnaire (MGH-SFQ) [55] and the Arizona Sexual Experiences Scale (ASEX) [56]. These multifaceted instruments quantify the diverse elements of sexual functioning in both women and men. A broader implementation of the MGH-SFQ, ASEX, and other multidimensional tools is crucial for advancing the scientific understanding of both female and male sexual well-being. Findings will inform the development of optimized, gender-inclusive models of care and interventions tailored to the complete spectrum of sexual issues and concerns and inform the provision of comprehensive patient-centered care. Lastly, it is important to note that this information highlights the need for a paradigm shift in the assessment of women’s sexual health. Moving beyond a narrow focus on sexual functioning and embracing multidimensional tools is not only a scientific imperative, but also a step toward providing comprehensive, patient-centered care. As the research community strives for a more inclusive understanding of sexual well-being, the incorporation of diverse perspectives and comprehensive measurement instruments will contribute to the development of more effective and empathetic models of care.

Regarding the focus on contextual, medical, or psychological areas, most of the studies explored the relationship of psychological and sexual health with a medical pathological origin, as mentioned above. In addition, it is important to highlight that, in some cases, the COVID-19 pandemic was an important factor when relating to the contextual area. Therefore, the effect of the COVID-19 pandemic was provided in the studies. In the work by Mistler et al. [28], most women reported no change in their sexual health behaviors, and very few reported an increase in sex-related behaviors due to COVID-19. During lockdown, sexually active subjects indicated lower levels of anxiety and depression [32]. Furthermore, other contextual aspects have been considered, for example, Polihronakis et al. [33] showed that discrimination in bisexual women has a direct relationship with internalization and sexual risk behaviors, increasing the odds of engaging in it and showing higher rates of STIs and HIV. To understand how discrimination against sexual minorities affects sexual health, more studies need to be conducted to better manage mental disorders in the LGBTQ+ community [57]. Lastly, psychosocial factors influence HSDD and interfere in the relationship with the partner and mental and emotional well-being [30]. More articles have shown that the problems that affect sexual health are medical in comparison to psychological issues. The review has highlighted the important role that medical aspects have in sexual health, compared to psychological and social aspects. A possible reason could be that the psychological and social aspects are still less relevant than the medical aspects.

Some limitations have been identified in this review. First, there is little attention to the exploration of positive dimensions within the domain of mental health. The prevailing evidence indicates that the articles under review predominantly adopted a clinical perspective in their treatment of psychological well-being, which mainly is the presence of depression and/or anxiety symptoms [38], or other mental health issues. The search provided studies that, in general, analyzed negative manifestations of mental health over the exploration and evaluation of positive aspects such as enjoyment or resilience. Second, the exclusion criteria of other type of documents (e.g., narrative and qualitative studies) and studies published in other languages limits the scope of information explored in this article. Other systematic reviews were not included in the analysis. Also, setting the age of the samples to 18 years of age could lose relevant aspects for the youngest women. Furthermore, only fully published studies have been included, which introduces the possibility of publication bias. Nevertheless, this systematic review offers a comprehensive overview of the association between PWB and sexual health in women.

## 5. Conclusions

In conclusion, through a meticulous analysis of the available research, this review provides information about valuable insights into the possible determinants of female sexual health. The association between PWB and female sexual health was demonstrated. The studies reviewed found that, from the female clinical samples, anxiety, depression, and general sexual functioning are the most relevant observed outcomes. In this line, the extensive prevalence and life impact of female sexual dysfunctions indicate these are critical public health issues warranting population-level interventions. Sexual health challenges affect a substantial portion of the female population and reduce quality of life [58]. The results of this systematic review call for implementing public health strategies beyond individual clinical approaches to comprehensively support women’s sexual well-being across their lifespans. In addition, the findings of this study will have significant implications for healthcare professionals and future research. Nevertheless, it is important to note that the limitations previously commented on could impact the generalizability and applicability of the findings. However, by recognizing the importance of psychological and sexual health issues, healthcare professionals will be able to be aware and attend to those issues, playing a crucial role in improving women’s sexual experiences, satisfaction, and overall quality of life. Also, there will be a pressing need to elevate female sexual and psychological wellness as a public health priority and develop policies, programs, and messaging to positively transform women’s experiences at a societal level. For example, the studies by Rezaei et al. [59] and Mahnza et al. [60] demonstrated the benefits of sexual health education programs for women. Rezaei et al. [59] implemented a program in Iran consisting of group sessions and individual consultations. They found positive impacts on female sexual function, notably desire and arousal, as well as improved sexual attitudes. These results align with Mahnza et al.’s findings [60] that sexual health education enhances women’s sexual function in general. Beyond improving sexual health, these programs may also prevent issues like unintended pregnancy, sexually transmitted infections, sexual abuse, and violence against women. Sexual health education equips women with the knowledge to make informed choices and build healthy relationships. Therefore, by taking a holistic perspective encompassing mental, emotional, and social well-being, researchers and healthcare providers can gain a deeper understanding of the multidimensional factors influencing women’s sexual health and develop impactful interventions to improve the lives of women across diverse communities. In addition, in the future, it would be recommended to develop patient-centered care and holistic models. Lastly, when conducting future complete assessments of psychological well-being, it is crucial to consider both positive mental health and mental health problems. The expanded perspective on mental health, which incorporates both positive mental health aspects and the absence of mental disorders, lends support to the recommendation of incorporating a broader range of positive psychology constructs and measures in future research. As a consequence, adopting this more holistic approach has the potential to improve the quality of future research and offer a more comprehensive understanding of the intricate relationship between psychological well-being and female sexuality.

## Figures and Tables

**Figure 1 healthcare-11-03025-f001:**
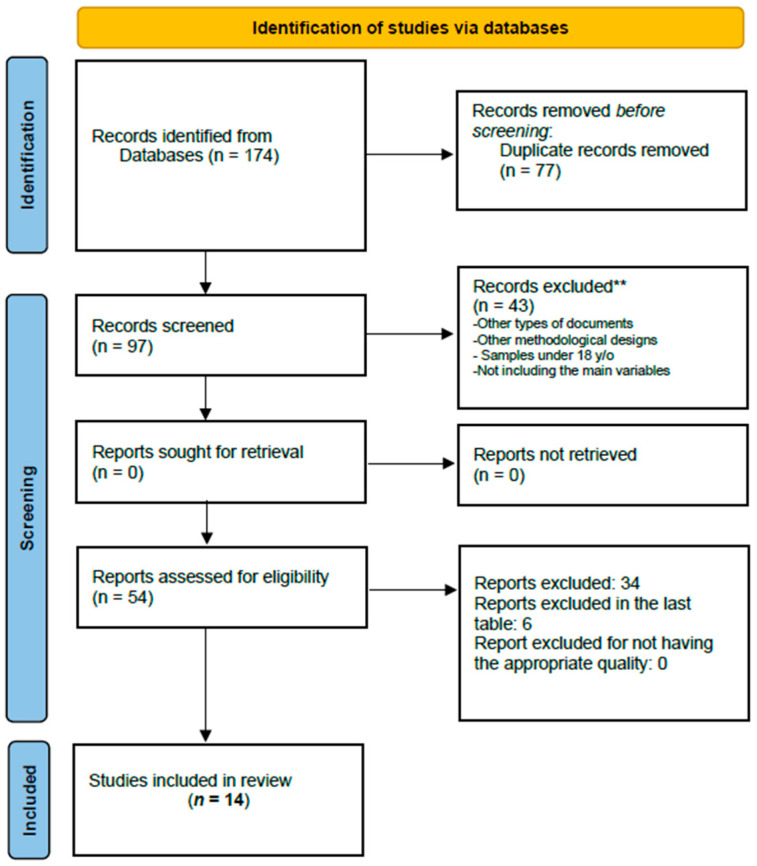
Flow diagram for the systematic review of searches of databases. ** Other types of documents, other methodological designs, samples under 18 y/o, not including the main variables.

**Table 1 healthcare-11-03025-t001:** Reviewed articles about sexual health and psychological well-being.

Study	Sample	Type of Sample			PWB Variable				Sexual Health Variable			Measures		Results		
		Clinical	Student	General	Anxiety	Depression	Other		Pertaining to Sexual Functioning	Other		Mental	Sexual	Type of Results	Type of Association	Understanding PWB
Cook et al. (2013) [36]	N = 180 adults with congenital heart disease with implantable cardioverter-defibrillators (ICDs) and without ICDs (44% females). n = 25 women with ICDs; n = 54 women without ICDs	x		x	x	x			x			The Florida Shock Anxiety Scale (FSAS), The Beck Depression Inventory-II (BDI-II)	Female Sexual Functioning Index (FSFI)	Psychological	Anxiety and depression were negatively associated with sexual functioning. A higher level of shock-related anxiety was associated with poorer sexual function in women.	PWB is understood and measured as the absence of anxiety and depression.
Dong et al. (2021) [25]	N = 1442 adults with infertility (57.4% females). n = 278 women with postponed fertility treatment; n = 549 women with fertility treatment not delayed	x			x		x	Quality of marriage	x			Generalized Anxiety Disorder scale (GAD-7), Patient Health Questionnaire (PHQ-9), Quality of Marriage Index (QMI)	Female Sexual Function Index (FSFI)	Contextual, Psychological	Postponed fertility treatment was directly associated with distress, and distress with sexual health. Delaying fertility treatment negatively affects sexual and psychological health.	PWB interpreted as psychological health included anxiety and depression symptoms and couple relations.
Dubin et al. (2020) [31]	N = 13,617 women			x			x	Self-confidence	x			Ad hoc 30-item online survey	Ad hoc 30-item online survey	Contextual	Partner’s ED was negatively associated with female psychological health, sexual satisfaction, and the success of the overall partnership.	PWB is considered as psychological factors included emotional (poor body image/self-esteem/stress and relational) marital or relationship problems.
Fogh et al. (2021) [26]	N = 333 women breast cancer survivors (BCSs)	x				x	x	Body image	x	x	Distress caused by sexual complain	Beck Depression Inventory (BDI), CARES (body image and relationship Satisfaction)	Female Sexual Function Index (FSFI), Sexual Complaint Screener—Women (SCS-W), ICIQ-FLUTSsex	Medical	PWB and relationship satisfaction were negatively associated with relevant sexual dysfunction (SD). Cancer treatment was associated with SD.	PWB is understood and measured as the absence of depression.
Hertz et al. (2022) [37]	N = 206 individuals (63.6% females). n = 89 with ADHD; n = 44 without ADHD	x		x			x	Attention deficit disorder	x	x	Sexual risk and Hypersexual behaviors	Self-Report Wender-Reimherr Adult Attention Deficit Disorder Scale (SR-WRAADDS)	Sexual Risk Survey (SRS), Hypersexual Behavior Inventory (HBI-19), Sexual Behavior Questionnaire-German Version (SBQ-G)	Medical, Psychological	Hypersexual behaviors, sexual risk taking, and SD were directly related to symptoms of emotional dysregulation, impulsivity, and oppositional symptoms in women with ADHD.	PWB is studied through domains attention difficulties, hyperactivity/restlessness, temper, affective lability, emotional over-reactivity, disorganization, and impulsivity.
Liñan-Bermudez et al. (2022) [27]	N = 60 women	x				x	x	Severity of climacteric symptoms	x	x	Severity of climacteric symptoms	Beck Depression Inventory (BDI), Menopause Rating Scale (3 domains: somatic, psychological, and urogenital)	Female Sexual Function Questionnaire-2 (FSM-2)	Medical	Urogenital aspects of climacteric were associated with depression and negatively associated with sexual functioning.	PWB is measured as the absence of depression mood, irritability, anxiety, and mental exhaustion.
Mistler et al. (2021) [28]	N = 110 individuals on methadone as treatment for OUD opioid use disorder (56% females)	x			x	x	x	Loneliness and frustration		x	Condomless sex and transactional sex behavior	Ad hoc measures about social and health. Indices of social, physical, and mental well-being, including substance use and mental health status	Ad hoc measures about sexual health behaviors (sexual hygiene, number of sexual partners, sex without condom, sexting, and transactional sex…)	Contextual, Psychological	There was no change in sexual-health-related behaviors but an increase in psychological distress, frustration or boredom, anxiety, depression, and loneliness due to COVID-19.	PWB is described as mental well-being including substance use, mental health status, and healthcare access, etc.
Mollaioli et al. (2021) [32]	N = 6821 individuals (61.24% females)			x	x	x	x	Dyadic adjustment	x			Generalized Anxiety Disorder scale (GAD-7), Patient Health Questionnaire (PHQ-9), Dyadic Adjustment Scale (DAS)	Orgasmometer (a single item about the intensity of the perception of the orgasmic experience), Female Sexual Function Index (FSFI)	Contextual, Psychological	Lack of sexual activity during COVID-19 confinement was directly associated with an increased risk of anxiety and depression.	PWB is addressed and analyzed as psychological health, specifically with anxiety and depression constructs.
Mooney et al. (2022) [38]	N = 152 partnered individuals (84.2% females) n = 76 couples with PGAD/GPD symptoms; n = 76 couples without PGAD/GPD symptons	x		x	x	x	x	Couple satisfaction	x	x	Sexual distress	PGAD/GPD Symptom Details, Couple-Satisfaction Index-Short Form (CSI), Hospital Anxiety and Depression Scale (HADS)	Global Measure of Sexual Satisfaction (GMSEX), Female Sexual Functioning Index (FSFI), Sexual Distress Scale (SDS)	Medical	PGAD/GPD symptoms were directly associated with low sexual and relationship satisfaction, increased sexual stress, and more symptoms of depression and anxiety.	PWB is “determined by the presence of depression and/or anxiety symptoms” ([38], p. 234).
Philip et al. (2013) [29]	N = 70 female rectal and anal cancer survivors	x			x	x	x	Quality of life and body image	x	x	Sexual enjoyment	Impact of Events Scale-Revised (IES-R), the Brief Symptom Inventory (BSI), Treatment of Cancer Core Quality of Life Questionnaire (EORTC-QLQ-C30)	Female sexual functioning index (FSFI), European Organization for Research and Treatment of Cancer Core Colorectal Cancer-Specific Module (EORTC-QLQ-CR38)	Medical	Body image, anxiety, and cancer-specific post-traumatic distress were negatively associated with sexual functioning.	PWB is addressed and analyzed with anxiety, depression, and body image variables and measures.
Polihronakis et al. (2021) [33]	N = 352 women			x			x	Attitudes toward appearance		x	Sexual objectification and anti-bisexual experiences, and sexual risk	Internalization-General (IG) subscale of the Sociocultural Attitudes Toward Appearance Questionnaire-3 (SATAQ-3), Objectified Body Consciousness Scale (OBCS-Survey)	Interpersonal Sexual Objectification Scale (ISOS), Anti-Bisexual Experiences Scale (ABES), Sexual Risk Survey (SRS)	Contextual, Psychological	Antibisexual discrimination is associated with lower psychological well-being. Discrimination in bisexual women has a direct relationship with internalization and sexual risk behaviors and higher rates of STIs and HIV.	PWB addressed as psychological functioning or psychological health.
Simon et al. (2022) [30]	N = 530 women with HSDD	x					x	Being satisfied with life, quality of sleep, mental ability, etc.	x			12-Item Short Form Survey (SF-12)	Female Sexual Functioning Index (FSFI)	Medical, Psychological	Symptoms of hypoactive sexual desire were associated with poor health in quality of life, mental well-being, and couple relationship.	PWB is described as mental and emotional well-being and assessed by measuring the degree of interference in emotional well-being, the ability to ‘‘stay in the moment,’’ satisfaction with life, being at peace with oneself, and feeling happy, among others.
Vedovo et al. (2021) [34]	N = 205 women. n = 125 transgender women; 80 cisgender women			x		x	x	Mental health and vitality	x			Beck Depression Inventory Primary Care (BDI-PC), General Health Survey (SF-36)	Female Sexual Function Index (FSFI), Operated Male to Female Sexual Function Index (OMtFSI)	Psychological	Sexual pain predicts the risk of depression in transgender people. The relationship between depressive symptoms and sexual function was greater in transgender people.	PWB is addressed with mental well-being variables and is measured with depression and mental health construct scales.
Vedovo et al. (2022) [35]	N = 2543 (43.4% female)			x		x	x	Mental health, vitality, and loneliness	x			Beck Depression Inventory Primary Care (BDI-PC), General Health Survey (SF-36), UCLA Loneliness Scale-version 3	Female Sexual Function Index (FSFI)	Contextual, Psychological	Female sexual function is associated with psychological variables such as anxiety and emotional satisfaction with relationships. Social constraints had a negative impact on female sexual function.	PWB is addressed and measured by depression, loneliness, and a generic health status scale. This latter evaluates physical, social, emotional, and medical health.

## Data Availability

Not applicable.

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
