# Peer review of "Sexual Health and Psychological Well-Being of Women: A Systematic Review"

_healthcare, 2023, doi:10.3390/healthcare11233025_

Round 1
Reviewer 1 Report
Comments and Suggestions for Authors
Congrats on your thorough work! Here are a few suggestions to improve the manuscript before it gets published:
Line 101 - Why did the authors limited article search between 2010-2023? Was there any specific reason for that choice?
Line 114 - The ratings on QATOCCS on the quality of papers should have been further described by providing means or percentages on quality criteria for the included studies and for identifying limitations.
Figure 1 - Number of included/excluded studies may not be correct? i.e., 54-34-5=15 studies instead of 14. Also, why the 34 studies were excluded and why the 5 studies were excluded in last table? Please provide reasons why.
I hope the above feedback helps.
Author Response
Dear Editor and Reviewers:
Thank you very much for giving us the opportunity to improve our manuscript entitled “Sexual Health and psychological well-being of women: A systematic review”. We highly appreciate all your valuable comments and recommendations, as well as the positive feedback received and the good evaluation of this review. We would like to thank you for your consideration of this manuscript in Healthcare. We have highlighted the changes in yellow within the revised document.
Best regards,
Reviewer 1
Congrats on your thorough work! Here are a few suggestions to improve the manuscript before it gets published: I hope the above feedback helps.
Response: Thank you very much. We highly appreciate your comments and recommendations.
Line 101 - Why did the authors limited article search between 2010-2023? Was there any specific reason for that choice?
Response: Thank you. The authors decided to review the most current information possible, so they decided to start from the last decade and include the last 3 years. We were interested in the latest updates and the latest information about it.
Line 114 - The ratings on QATOCCS on the quality of papers should have been further described by providing means or percentages on quality criteria for the included studies and for identifying limitations.
Response: Thank you. The information about QATOCCS has been revised. It is a checklist, the items considered most appropriate to the type of article were chosen. For the current analysis, the two reviewers independently read and rated each of the selected studies across the QATOCCS criteria. This comprehensive quality assessment ensured that only methodologically robust studies with reliable psychometrics and minimal bias were included, strengthening the reliability of results. There was no scoring that can be provided, the results from the checklist were qualitative, not numerical.
Figure 1 - Number of included/excluded studies may not be correct? i.e., 54-34-5=15 studies instead of 14. Also, why the 34 studies were excluded and why the 5 studies were excluded in last table? Please provide reasons why.
Response: Thank you very much for the suggestion. It was a typo; it has already been corrected. There were 6 reports excluded in the last table.
Reviewer 2 Report
Comments and Suggestions for Authors
Respected Authors
Thank you for addressing an important topic about Sexual health and psychological well-being among women. Your article is generally good, but in some cases it needs corrections.
- Please remove the subheading from your abstract.
- Line 13, add "PRISMA checklist" or guideline.
- Abstract, please add the quality appraisal checklist and the methods of synthesis.
- Line 101, please add your rationale for selecting this period for search.
- Based on your PRISMA, please add the "selection process" section to your study (PRISMA 2020, item 9).
- Line 110, please add all items that you extracted from the included studies.
- Eligibility criteria, please add the design of the included studies.
- Please provide more details about the quality assessment tool. Including the number of items and scoring method.
- Line 132, please remove Table S1 from the methods section, this table is related to your results. Please consider items 13 a-f and add the related methods of synthesis.
- Please restructure your result considering PRISMA items (PRISMA 2020, items 16a-20).
- Please add Table S1 in the results section, at the end of the "study characteristics" section.
Thanks
Author Response
Dear Editor and Reviewers:
Thank you very much for giving us the opportunity to improve our manuscript entitled “Sexual Health and psychological well-being of women: A systematic review”. We highly appreciate all your valuable comments and recommendations, as well as the positive feedback received and the good evaluation of this review. We would like to thank you for your consideration of this manuscript in Healthcare. We have highlighted the changes in yellow within the revised document.
Best regards,
Reviewer 2
Respected Authors
Thank you for addressing an important topic about Sexual health and psychological well-being among women. Your article is generally good, but in some cases it needs corrections.
Response: Thank you very much. We highly appreciate your comments and recommendations.
- Please remove the subheading from your abstract.
Response: Thank you. The abstract follows the guidelines of the journal. There is no subheading in the last version of the manuscript.
- Line 13, add "PRISMA checklist" or guideline.
Response: Thank you for your comment. It is already done.
- Abstract, please add the quality appraisal checklist and the methods of synthesis.
Response: Thank you for your comment. It is already done.
- Line 101, please add your rationale for selecting this period for search.
Response: Thank you. The authors decided to review the most current information possible, so they decided to start from the last decade and including the last 3 years. We were interested in the latest updates and the latest information about it.
- Based on your PRISMA, please add the "selection process" section to your study (PRISMA 2020, item 9).
Response: Thank you very much for your recommendation. The selection process has been added.
- Line 110, please add all items that you extracted from the included studies.
Response: Thank you. This information is in the Synthesis Methods paragraph.
- Eligibility criteria, please add the design of the included studies.
Response: Thank you very much for your recommendation. This information has been added.
- Please provide more details about the quality assessment tool. Including the number of items and scoring method.
Response: Thank you. The information about QATOCCS has been revised. It is a checklist, the items considered most appropriate to the type of article were chosen. For the current analysis, the two reviewers independently read and rated each of the selected studies across the QATOCCS criteria. This comprehensive quality assessment ensured that only methodologically robust studies with reliable psychometrics and minimal bias were included, strengthening the reliability of results. There was no scoring that can be provided, the results from the checklist were qualitative, not numerical.
- Line 132, please remove Table S1 from the methods section, this table is related to your results. Please consider items 13 a-f and add the related methods of synthesis.
Response: Thank you very much. We have removed Table S1 from the methods section. Information about methods of synthesis has already been provided.
- Please restructure your result considering PRISMA items (PRISMA 2020, items 16a-20).
Response: Thank you very much for your recommendations. This section has been revised. We have considered PRISMA items for the paragraphs of our results section.
- Please add Table S1 in the results section, at the end of the "study characteristics" section.
Response: Thank you very much. We have added Table S1 in the results section.
Reviewer 3 Report
Comments and Suggestions for Authors
The paper entitled “Sexual health and psychological well-being of women: A systematic review” aimed to fill a critical knowledge gap by examining existing research on the association between PWB and sexual health in women, paving the way for more effective strategies to enhance women's lives. The intersection of psychological well-being (PWB) and female sexual health is a crucial yet understudied area with significant implications for women's overall quality of life. Understanding the intricate relationship between these two aspects not only addresses the pressing issue of sexual dysfunction but also offers insights into promoting holistic well-being for women. This is a well-written article, although there are some concerns about several issues that the authors should properly address before considering the manuscript for publication in Healthcare.
INTRODUCTION:
The introduction provides a comprehensive overview of the importance of psychological well-being (PWB) and female sexual health and their interrelationship. It effectively highlights the significance of understanding this complex interplay for women's overall quality of life and well-being. However, there is a need for further clarification regarding the research objectives to enhance the focus of this systematic review.
The introduction introduces two primary aspects: risk factors associated with female sexual dysfunction and the assessment tools used to measure sexual health. While both these aspects are undoubtedly crucial, the current presentation of mixed objectives may lead to some confusion. To improve the clarity and focus of the review, it is advisable to either prioritize one aspect as the primary objective or clearly delineate separate objectives for risk factors and assessment tools. Additionally, providing a specific research question or hypothesis that guides the systematic review is crucial. A well-defined research question will help readers understand the scope and purpose of the review more clearly. As systematic reviews are considered a high form of research evidence, a precise research question is essential to ensure that the review's objectives are met efficiently.
In summary, while the introduction effectively establishes the importance of the topic, it would greatly benefit from a more explicit and focused research question or objective that clarifies whether the review will primarily address risk factors, assessment tools, or both. This clarity will enhance the overall quality and impact of the systematic review.
METHODS
The Methods section effectively outlines the systematic review process, adhering to PRISMA guidelines and ensuring methodological transparency. The inclusion and exclusion criteria are well-defined, and the search strategy appears thorough. The use of the Quality Assessment Tool for Observational Cohort and Cross-Sectional Studies (QATOCCS) to assess study risk of bias is commendable, enhancing the reliability of the selected studies. The tabulated results in Table S1 offer a concise summary of key study details. Overall, this section demonstrates a rigorous and systematic approach to data collection and synthesis. However, I have a question regarding the presentation of results.
Could you please clarify the rationale for presenting the results in a supplementary table rather than including key findings in the main text? This clarification would help ensure that the main findings are readily accessible to readers and enhance the overall impact of your research.
RESULTS
In the Results section, the authors have effectively summarized the selection process and characteristics of the included studies using a flowchart (e.g., PRISMA flowchart). However, it is essential to explicitly indicate the reasons for exclusions at each stage of the flowchart to provide readers with a more transparent understanding of the study selection process. This will help clarify why certain studies were excluded, enhancing the overall clarity of the article.
DISCUSSION
In the Discussion section, the authors provide a comprehensive overview of the findings and implications of their systematic review. They effectively highlight the predominant focus on women with clinical conditions and the prevalence of anxiety and depression as key mental health variables in the reviewed studies. The discussion on the measurement instruments used for assessing sexual functioning, particularly the FSFI, and the recommendation for broader implementation of multidimensional tools like the MGH-SFQ and ASEX is insightful. The authors also acknowledge the impact of contextual factors, such as the COVID-19 pandemic, on women's sexual health.
However, it would be beneficial for the authors to further elaborate on the potential reasons behind the observed focus on medical aspects over psychological and social ones in the reviewed studies. Additionally, discussing the implications of these findings for future research directions and the development of holistic, patient-centered care models would enhance the conclusion of the manuscript.
Lastly, while the authors acknowledge some limitations of their review, it might be useful to briefly discuss the implications of these limitations on the interpretation of the results and suggest avenues for future research that could address these limitations.
CONCLUSION
The Conclusion section provides a well-structured summary of the systematic review's key findings and their implications. It effectively underscores the importance of recognizing the association between psychological well-being and female sexual health. The call for population-level interventions and elevating female sexual and mental wellness as a public health priority is commendable. The reference to studies on sexual health education programs highlights the potential for actionable solutions.
However, it would be beneficial for the authors to briefly discuss the limitations of their review once again in the conclusion and emphasize how these limitations may impact the generalizability and applicability of their findings. Additionally, offering some final thoughts on the broader implications of their work for clinical practice, policy development, and future research could enhance the conclusion.
Author Response
Dear Editor and Reviewers:
Thank you very much for giving us the opportunity to improve our manuscript entitled “Sexual Health and psychological well-being of women: A systematic review”. We highly appreciate all your valuable comments and recommendations, as well as the positive feedback received and the good evaluation of this review. We would like to thank you for your consideration of this manuscript in Healthcare. We have highlighted the changes in yellow within the revised document.
Best regards,
Reviewer 3
The paper entitled “Sexual health and psychological well-being of women: A systematic review” aimed to fill a critical knowledge gap by examining existing research on the association between PWB and sexual health in women, paving the way for more effective strategies to enhance women's lives. The intersection of psychological well-being (PWB) and female sexual health is a crucial yet understudied area with significant implications for women's overall quality of life. Understanding the intricate relationship between these two aspects not only addresses the pressing issue of sexual dysfunction but also offers insights into promoting holistic well-being for women. This is a well-written article, although there are some concerns about several issues that the authors should properly address before considering the manuscript for publication in Healthcare.
Response: Thank you very much. We highly appreciate your comments and recommendations.
INTRODUCTION:
The introduction provides a comprehensive overview of the importance of psychological well-being (PWB) and female sexual health and their interrelationship. It effectively highlights the significance of understanding this complex interplay for women's overall quality of life and well-being. However, there is a need for further clarification regarding the research objectives to enhance the focus of this systematic review.
The introduction introduces two primary aspects: risk factors associated with female sexual dysfunction and the assessment tools used to measure sexual health. While both these aspects are undoubtedly crucial, the current presentation of mixed objectives may lead to some confusion. To improve the clarity and focus of the review, it is advisable to either prioritize one aspect as the primary objective or clearly delineate separate objectives for risk factors and assessment tools. Additionally, providing a specific research question or hypothesis that guides the systematic review is crucial. A well-defined research question will help readers understand the scope and purpose of the review more clearly. As systematic reviews are considered a high form of research evidence, a precise research question is essential to ensure that the review's objectives are met efficiently.
In summary, while the introduction effectively establishes the importance of the topic, it would greatly benefit from a more explicit and focused research question or objective that clarifies whether the review will primarily address risk factors, assessment tools, or both. This clarity will enhance the overall quality and impact of the systematic review.
Response: Thank you very much for your recommendation. We agree with you. We have deeply revised the introduction and added more information. The research question has been included.
METHODS
The Methods section effectively outlines the systematic review process, adhering to PRISMA guidelines and ensuring methodological transparency. The inclusion and exclusion criteria are well-defined, and the search strategy appears thorough. The use of the Quality Assessment Tool for Observational Cohort and Cross-Sectional Studies (QATOCCS) to assess study risk of bias is commendable, enhancing the reliability of the selected studies. The tabulated results in Table S1 offer a concise summary of key study details. Overall, this section demonstrates a rigorous and systematic approach to data collection and synthesis. However, I have a question regarding the presentation of results.
Could you please clarify the rationale for presenting the results in a supplementary table rather than including key findings in the main text? This clarification would help ensure that the main findings are readily accessible to readers and enhance the overall impact of your research.
Response: Thank you very much. Instruction for authors of this journal encourages the supplementary material published online alongside the manuscript (Table S1).
RESULTS
In the Results section, the authors have effectively summarized the selection process and characteristics of the included studies using a flowchart (e.g., PRISMA flowchart). However, it is essential to explicitly indicate the reasons for exclusions at each stage of the flowchart to provide readers with a more transparent understanding of the study selection process. This will help clarify why certain studies were excluded, enhancing the overall clarity of the article.
Response: Thank you very much for the recommendation. The PRISMA flowchart has been revised. The reasons for exclusions have been provided in the flowchart.
DISCUSSION
In the Discussion section, the authors provide a comprehensive overview of the findings and implications of their systematic review. They effectively highlight the predominant focus on women with clinical conditions and the prevalence of anxiety and depression as key mental health variables in the reviewed studies. The discussion on the measurement instruments used for assessing sexual functioning, particularly the FSFI, and the recommendation for broader implementation of multidimensional tools like the MGH-SFQ and ASEX is insightful. The authors also acknowledge the impact of contextual factors, such as the COVID-19 pandemic, on women's sexual health.
However, it would be beneficial for the authors to further elaborate on the potential reasons behind the observed focus on medical aspects over psychological and social ones in the reviewed studies. Additionally, discussing the implications of these findings for future research directions and the development of holistic, patient-centered care models would enhance the conclusion of the manuscript.
Lastly, while the authors acknowledge some limitations of their review, it might be useful to briefly discuss the implications of these limitations on the interpretation of the results and suggest avenues for future research that could address these limitations.
Response: Thank you very much for your recommendation. We have deeply revised the discussion and conclusion of the manuscript. We have enhanced these sections.
CONCLUSION
The Conclusion section provides a well-structured summary of the systematic review's key findings and their implications. It effectively underscores the importance of recognizing the association between psychological well-being and female sexual health. The call for population-level interventions and elevating female sexual and mental wellness as a public health priority is commendable. The reference to studies on sexual health education programs highlights the potential for actionable solutions.
However, it would be beneficial for the authors to briefly discuss the limitations of their review once again in the conclusion and emphasize how these limitations may impact the generalizability and applicability of their findings. Additionally, offering some final thoughts on the broader implications of their work for clinical practice, policy development, and future research could enhance the conclusion.
Response: Thank you very much. We have deeply revised the discussion and conclusion of the manuscript. Information about limitations, future research, as well as work for clinical practice has been provided.
Reviewer 4 Report
Comments and Suggestions for Authors
Thank you for the opportunity to review your paper. Please refer to the attached file for my review.

The manuscript is mostly well written but requires clarification/correction in various sections. In my attached review I have listed errors or points requiring clarification/correction that I identified while reading.
Author Response
Dear Editor and Reviewers:
Thank you very much for giving us the opportunity to improve our manuscript entitled “Sexual Health and psychological well-being of women: A systematic review”. We highly appreciate all your valuable comments and recommendations, as well as the positive feedback received and the good evaluation of this review. We would like to thank you for your consideration of this manuscript in Healthcare. We have highlighted the changes in yellow within the revised document.
Best regards,
Reviewer 4
Thank you for the opportunity to review your manuscript, which details a systematic review of literature published during the period 2010-2023 and reporting studies exploring associations between psychological wellbeing (PSB) and adult women’s sexual health. The review as reported was performed according to suitable standards and I have no issue with the way the review was done.
Response: Thank you very much. We highly appreciate your comments and recommendations.
However, I do query if the authors are intending to infer that PSB and mental health are synonymous. For instance, the search string very clearly indicates a focus on PSB: Lines 100-102. “The search strategy was [“Sexual health” AND “Psychological well-being” AND (“women” OR “female”)].” However, the eligibility criteria (Lines 88-89) suggest otherwise: “Studies that evaluated sexual health and mental health or PWB variables. Studies that associated sexual health and mental health or PWB variables.” There are also several mentions of literature concerning mental health. For example, in the abstract: Lines 13-14. “The inclusion criteria were studies with samples of adult women, that evaluated and associated sexual functioning and mental health.” Line 16. “variables related to mental and sexual health were examined”. Lines 18-19. “focused on anxiety and depression as main mental health variables”. Lines 68-72. “Despite the significance of this topic, more research is needed to identify risk factors for sexual dysfunction and better understand the interaction between PWB and female sexual health. Lastly, the examination of recent studies with certain populations, and informing the most used measures that assess women's sexual health and mental health are issues that need to be addressed in research.” Lines 76-78. “Thus, the aim of this study is to conduct a comprehensive review of the existing literature to explore the association between PWB and sexual health in women.” There are various other references to mental health throughout. While it’s mostly accepted that PBS and mental health are related concepts, it’s also accepted that these concepts are not interchangeable. The citations used for the initial definition of PSB are all Ryff, who herself argues against considering these concepts synonymously: “To recapitulate, it is now well established that eudaimonic well-being is not simply the flipside of psychological distress. Both are important indicators of overall mental health, and population studies reveal diverse combinations of how the two domains come together.” https://doi.org/10.1159/000353263 From my reading of Ryff’s work over the years, it seems that the concept of PSB relates to positive functioning, and this focus emerged as a reaction against the focus on mental health as negative or disordered functioning. (e.g., see Ryff & Singer, Psychological Well-Being: Meaning, Measurement, and Implications for Psychotherapy Research. Psychother Psychosom (1996) 65 (1): 14–23). It is therefore important to clearly differentiate a focus on PSB OR on mental health or otherwise clarify that the focus was on both PSB AND mental health and include some suitable differentiating descriptions to acknowledge that you’re not using the terms interchangeably. The manuscript is mostly well written but requires clarification/correction in various sections.
Response: Thank you very much for your comments. We agree with you. We have revised, modified and clarified these terms throughout the document.
Here are errors or points requiring clarification/correction I identified while reading. Line 108. ‘screed’. Should this be ‘screened’?
Response: Thank you. In the document, line 108, it is written “screened”.
Lines 197-199. “This section may be divided by subheadings. It should provide a concise and precise description of the experimental results, their interpretation, as well as the experimental conclusions that can be drawn.” These statements appear to be editorial advice provided in the journal’s template; should have been deleted.
Response: Thank you. These statements have been deleted.
Lines 214-216. “Some psychological disorders like Attention-Deficit/Hyperactivity Disorder (ADHD) and opioid use disorder are also considered, for example, women with ADHD reported more hypersexual behaviors [37].” Split into two with new sentence beginning at ‘For example…’.
Response: Thank you. Done.
Line 264. “It has been highlighted the important role that medical aspects have…”. Reword for clarity: “The review has highlighted the important role that medical aspects have…”
Response: Thank you. Done.
Line 272. “Furthermore, only fully published studies have been included which possibility of publication bias.” Reword for clarity: “Furthermore, only fully published studies have been included, which introduces the possibility of publication bias.”
Response: Thank you. Done.
Lines 280-281. “…that female clinical samples, anxiety and depression, and general sexual functioning are the most relevant observed outcomes.” Reword for clarity; clinical samples are not an outcome.
Response: Thank you. Done.
Reference list. Entries appear to be misaligned, with item 1 being absent and item entries beginning with number 2.
Response: Thank you. Done.
Table S1. Title. “Revised articles about sexual health and psychological well-being”. Should this be ‘Reviewed articles…’?
Response: Thank you. Done.
Table S1. The use of white space can be enhanced with prudent minimising of column widths in columns with only ‘x’ or blank entries (e.g., Type of sample, Mental health variable except Other, and Sexual health variable including Other). Letters could be used as column headers with a note inserted beneath the table to indicate what the letters correspond to. This could allow for columns with lots of text to be broadened somewhat. Single spacing would also enhance the readability of the table. The Sample and Type of results column widths could also be reduced to gain column space in columns with more text. Table S1. There are 2 typos: Psychologycal (pp. 4 & 9 of the table). Table S1. Pagination on these pages is odd; indicates pp. 1-10 ‘of 20’.
Response: Thank you. The table format is according to the edition of the journal.
Round 2
Reviewer 3 Report
Comments and Suggestions for Authors
I would like to express my appreciation for your prompt response and the revisions made to the manuscript titled "Sexual Health and Psychological Well-being of Women: A Systematic Review." Your efforts to address my previous comments are noted and valued. I have reviewed the revised manuscript and would like to provide further feedback.
While I acknowledge the improvements made in several sections of the paper, there remains a fundamental concern regarding the research objective. The central issue is the need for greater clarity regarding the objective of the systematic review. In my previous review, I emphasized the importance of specifying the research question or hypothesis and specifying the type of association (e.g., odds ratios, prevalence ratios, etc.) that the paper aims to explore. Regrettably, this critical aspect has not been adequately addressed in the revised manuscript.
It should be noted that a well-defined research question or objective is essential to guide a systematic review effectively. It not only helps readers understand the scope and purpose of the study, but also ensures that the objectives are met efficiently. Therefore, I urge you to reconsider and clearly specify the research objective of your paper in the introduction section. This will enhance the overall quality and impact of your systematic review.
Additionally, I recommend you indicate the type of association in the table with the main results. This will provide readers with a quick and clear reference to the nature of the associations explored in the study. This clarification is essential to ensure that the paper is easily understandable and beneficial to the target audience.
I understand these revisions may require some adjustments to the introduction and the presentation of results, but I believe that such clarification will significantly strengthen the manuscript and better serve your readers.
Once these revisions have been made, I would be more than willing to re-evaluate the manuscript to ensure that it effectively addresses these critical concerns.
Author Response
I would like to express my appreciation for your prompt response and the revisions made to the manuscript titled "Sexual Health and Psychological Well-being of Women: A Systematic Review." Your efforts to address my previous comments are noted and valued. I have reviewed the revised manuscript and would like to provide further feedback.
While I acknowledge the improvements made in several sections of the paper, there remains a fundamental concern regarding the research objective. The central issue is the need for greater clarity regarding the objective of the systematic review. In my previous review, I emphasized the importance of specifying the research question or hypothesis and specifying the type of association (e.g., odds ratios, prevalence ratios, etc.) that the paper aims to explore. Regrettably, this critical aspect has not been adequately addressed in the revised manuscript. It should be noted that a well-defined research question or objective is essential to guide a systematic review effectively. It not only helps readers understand the scope and purpose of the study, but also ensures that the objectives are met efficiently. Therefore, I urge you to reconsider and clearly specify the research objective of your paper in the introduction section. This will enhance the overall quality and impact of your systematic review.
Response: Thank you very much. We highly appreciate your comments and have tried to follow them. Odds ratios reported for some of the reviewed and cited studies have been included in the text. Unfortunately, there were few studies that provided specifically this information (i.e., OR). Furthermore, as specified at the end of the introduction section, this work started with the research question “How is psychological well-being related to sexual health in women?”. Based on previous research and our expertise, we hypothesized that better psychological well-being would be related to better female sexual health. Thus, the main objective was to conduct a comprehensive review of the existing scientific literature to explore the association between PWB and sexual health in women.
Additionally, I recommend you indicate the type of association in the table with the main results. This will provide readers with a quick and clear reference to the nature of the associations explored in the study. This clarification is essential to ensure that the paper is easily understandable and beneficial to the target audience.
Response: Thank you very much. This information has been added to the table.
I understand these revisions may require some adjustments to the introduction and the presentation of results, but I believe that such clarification will significantly strengthen the manuscript and better serve your readers.
Once these revisions have been made, I would be more than willing to re-evaluate the manuscript to ensure that it effectively addresses these critical concerns.
Response: Thank you very much. We highly appreciate all your comments and recommendations.
Reviewer 4 Report
Comments and Suggestions for Authors
Thank you for your responses and the revised manuscript. Unfortunately, the revision now confirms to me that there is a major problem with the way the systematic review was performed, given the focus in the results on mental health and the absence of any reporting on PWB. I have attached further comments for your consideration.

The quality of English language expression is not the problem with this paper.
Author Response
Thank you for your responses and the revised manuscript. Unfortunately, the revision now confirms to me that there is a major problem with the way the systematic review was performed, given the focus in the results on mental health and the absence of any reporting on PWB. I have attached further comments for your consideration.
Response: Thank you very much. We highly appreciate all your comments. We agree with you and were concerned about this matter while we were writing the manuscript. After reviewing all the references included in this manuscript, it has been proved that psychological well-being (PWB) is considered a synonym for, or it is approached from the perspective of, psychological health, mental health, mental well-being, or the absence of a mental diagnosis (anxiety and/or depression), among others (please see Mooney et al., 2022 and/or Woloski-Wruble et al., 2010). There are studies that refer to PWB and mental health interchangeably. We have added information about this fact in the text. Furthermore, we have included in the table a new column for clarification.
Abstract.
Lines 18-21. “Results: 14 selected articles were analyzed, in which population samples and variables related to mental and sexual health were examined. 42.9% of the studies included clinical samples, 71.4% focused on anxiety and depression as main mental health variables, and 50% examined female sexual functioning as a sexual health variable.”
Response: Thank you. This sentence has been reviewed.
- The study focus is purportedly on PWB and yet there is no mention of PWB in the statements of Results. To the contrary, results refer to mental health and mental health variables such as anxiety and depression. I refer to my previous comments concerning the problematisation in referring to PWB and mental health interchangeably. Even if mental health emerged from the search, the focus of the results reporting should be on the search strategy (i.e., [“Sexual health” AND “Psychological well-being” AND (“women” OR “female”)]).
Response: Thank you very much. The statements of Results have been reviewed. This fact has also been added as a limitation of the study (”…many synonyms to refer to PWB emerged from the search, in some studies, it is only determined as the presence of depression and/or anxiety symptoms”).
Line 115. “The two reviewers screed the abstracts separately.” I raised this spelling error previously. It should be ‘screened’ not ‘screed’.
Response: Thank you. This has been corrected in the text.
Lines 145-147. “Thus, in the present systematic review study, 14 selected articles were analyzed, in which population samples and variables related to mental and sexual health were examined.” Again, reference to mental health instead of PWB. Based on the repeated reference to mental health instead of PWB, I query the accuracy of the screening by the two reviewers. It isn’t enough to simply make changes to the text throughout; the review procedure should be followed again to ensure that all 174 articles are assessed according to the search criteria and study aim. The entire results section comes into question due to the focus on mental health instead of PWB.
Response: Thank you again. As previously commented, we have reviewed all the references included in this manuscript in order to be sure that the articles are in accordance with the search criteria and the study's main aim. Furthermore, this sentence has been reviewed: (…variables related to psychological and sexual health were examined).
A search for literature on PWB should result in studies reporting, for example, hedonic (enjoyment, pleasure) and eudaimonic (meaning, fulfillment) happiness, as well as resilience (coping, emotion regulation, healthy problem solving). Instead, results reported in the review refer to mental health repeatedly, and facets of mental health such as anxiety, depression, and ADHD. The only occurrence of ‘happiness’ in your entire paper is in your reference list [2]. ‘Satisfaction’ occurs 18 times but that’s mostly in relation to sexual satisfaction. ‘Coping’ occurs once, in the reference list [57]. Other PWB descriptors do not feature at all.
Response: Thank you very much. We agree with you. Surprisingly, the search provided studies with negative facets of mental health rather than other more positive ones such as enjoyment or coping. This statement has been added as a limitation. Nevertheless, as previously commented, it has been observed that psychological well-being (PWB) is considered a synonym for, or is addressed under the perspective of, psychological health, mental health, mental well-being, or the absence of a mental diagnosis (mainly, anxiety and/or depression), among others.
Further, there is reference to “measures used to assess variables related to mental and sexual health” such as the Florida Shock Anxiety Scale, the Generalized Anxiety Disorder Scale, the Beck Depression Inventory, and the Hospital Anxiety and Depression Scale. There is no reference to measures of PWB.
Response: Thank you. This sentence has been reviewed. The search reported that most of the measures to assess variables related to psychological aspects focused on anxiety or depression. Furthermore, some studies explicitly mention that PWB was measured with anxiety or depression measures (see Mooney et al., 2020 or Philipo et al., 2013).
The review should be performed again with tighter conceptualisation of the search strategy and aligned screening procedure to ensure that the final set of papers can address the proposed question/s. If the intention is to include mental health as well as PWB, you need to revise your search string to include that. I find it odd that a systematic search for literature on PWB has not resulted in any studies that measured PWB but rather measured mental health. It seems to me that the review hasn’t been performed correctly after al.
Response: Thank you very much. As previously commented, we were concerned about this matter while we were writing the manuscript. We have tried to clarify all the possible misunderstandings within the manuscript. We have carefully revised the articles and synonyms for PWB, as well as clarified them in the text. We believe that the manuscript has now been improved.